# *KMT2A*: Umbrella Gene for Multiple Diseases

**DOI:** 10.3390/genes13030514

**Published:** 2022-03-15

**Authors:** Silvia Castiglioni, Elisabetta Di Fede, Clara Bernardelli, Antonella Lettieri, Chiara Parodi, Paolo Grazioli, Elisa Adele Colombo, Silvia Ancona, Donatella Milani, Emerenziana Ottaviano, Elisa Borghi, Valentina Massa, Filippo Ghelma, Aglaia Vignoli, Elena Lesma, Cristina Gervasini

**Affiliations:** 1Department of Health Sciences, Università Degli Studi di Milano, 20142 Milan, Italy; silvia.castiglioni1@unimi.it (S.C.); elisabetta.difede@unimi.it (E.D.F.); clara.bernardelli@unimi.it (C.B.); antonella.lettieri@unimi.it (A.L.); chiara.parodi@unimi.it (C.P.); paolo.grazioli@unimi.it (P.G.); elisaadele.colombo@unimi.it (E.A.C.); silvia.ancona@unimi.it (S.A.); emerenziana.ottaviano@unimi.it (E.O.); elisa.borghi@unimi.it (E.B.); valentina.massa@unimi.it (V.M.); filippo.ghelma@unimi.it (F.G.); aglaia.vignoli@unimi.it (A.V.); elena.lesma@unimi.it (E.L.); 2“Aldo Ravelli” Center for Neurotechnology and Experimental Brain Therapeutics, Università Degli Studi di Milano, 20142 Milan, Italy; 3Pediatric Highly Intensive Care Unit, Department of Pathophysiology and Transplantation, Università Degli Studi di Milano, Fondazione IRCCS Ca’ Granda Ospedale Maggiore Policlinico, 20122 Milan, Italy; donatella.milani@policlinico.mi.it; 4Child NeuroPsychiatry Unit, ASST Grande Ospedale Metropolitano Niguarda, 20162 Milan, Italy

**Keywords:** *KMT2A*, chromatinopathies, tumors, epigenetics

## Abstract

*KMT2A* (Lysine methyltransferase 2A) is a member of the epigenetic machinery, encoding a lysine methyltransferase responsible for the transcriptional activation through lysine 4 of histone 3 (H3K4) methylation. *KMT2A* has a crucial role in gene expression, thus it is associated to pathological conditions when found mutated. *KMT2A* germinal mutations are associated to Wiedemann–Steiner syndrome and also in patients with initial clinical diagnosis of several other chromatinopathies (i.e., Coffin–Siris syndromes, Kabuki syndrome, Cornelia De Lange syndrome, Rubinstein–Taybi syndrome), sharing an overlapping phenotype. On the other hand, *KMT2A* somatic mutations have been reported in several tumors, mainly blood malignancies. Due to its evolutionary conservation, the role of KMT2A in embryonic development, hematopoiesis and neurodevelopment has been explored in different animal models, and in recent decades, epigenetic treatments for disorders linked to KMT2A dysfunction have been extensively investigated. To note, pharmaceutical compounds acting on tumors characterized by *KMT2A* mutations have been formulated, and even nutritional interventions for chromatinopathies have become the object of study due to the role of microbiota in epigenetic regulation.

## 1. Introduction

*KMT2A* (Lysine methyltransferase 2A), also known as *MLL1,* is a protein coding gene mapping to human chromosome 11 (11q23.3), made up of 90,343 bases (GRCh38/hg38) and 37 exons belonging to KMTs (Lysine methyltransferases) family.

KMTs catalyze the transfer of methyl groups from S-adenosylmethionine to the lysine residues on histone tails, particularly the histone H3 tail. Unlike other epigenetic enzymes such as acetyltransferases (HATs), KMTs are more specific and usually modify one or two lysines on a single histone [1]. Lysines can be monomethylated, bimethylated or trimethylated without changing the electric charge of the amino acid side chain. The effect on chromatin state, i.e., whether it activates transcription or represses it, depends on the methylation states and their positions (Figure 1) [2,3,4,5,6,7,8,9,10,11,12,13,14,15]. KMTs are so called writers, enzymes that catalyze the addition of chemical groups to histone tails or to DNA; these modifications are not permanent but can be removed by erasers to reverse the influence on gene expression. Readers possess specialized domains able to recognize and interpret different chemical modifications. Writers, erasers and readers form the epigenetic machinery, and mutations in genes coding for this apparatus lead to ann altered chromatin conformation and an incorrect gene expression, resulting in a series of syndromes known as chromatinopathies, Mendelian genetic diseases, most of them with a dominant character [16,17,18]. Pathogenic mutations in KMTs and KDMs (Lysine demethylases) lead to haploinsufficiency in numerous developmental syndromes (Figure 2) (Table 1) [10,19].

Many species have a *KMT2A* ortholog, including fishes, birds, amphibians, and mammals; thus, its evolutionary conservation allowed a comprehensive study of KMT2A molecular functions through in vivo experiments on animal models (*Drosophila melanogaster*, *Danio rerio*, *Mus musculus*). *KMT2A* expression is mainly nuclear and ubiquitously present in 27 tissues, especially in ovary, lymph node, endometrium, thyroid and brain tissue [20]. *KMT2A* encodes a lysine methyltransferase (KMT) formed of 3969 amino acids, a transcriptional co-activator which plays a crucial role in hematopoiesis, in regulating gene expression at early developmental stages, and in the control of circadian gene expression. KMT2A is processed by the endopeptidase Taspase 1 in two fragments (MLL-C and MLL-N) which heterodimerize and regulate the transcription of specific genes, including *HOX* genes [21]. KMT2A protein has 18 domains, including the CXXC-type zinc finger, the extended PHD domain and the bromodomain. The SET domain has the methyltransferase activity (mono-, di-, tri-methylation) on lysine 4 of histone 3 (H3K4 me1/2/3), a post-transcriptional modification (PTM) responsible of epigenetic transcriptional activation and which efficiency can be increased when the protein is associated with another component of the MLL1/MLL complex (Figure 3) [22].

As other members of KMTs family, KMT2A regulates gene transcription through chromatin opening or closure and its activity is antagonized by the lysine demethylases (KDMs) family.

## 2. *KMT2A* Germline Mutations

### 2.1. Wiedemann–Steiner Syndrome

*KMT2A* germinal variants are associated to the Wiedemann–Steiner syndrome (WDSTS, OMIM #605130), a rare autosomal dominant disorder characterized by different features, mainly intellectual disability (ID), developmental delay (DD), pre- and post-natal growth deficiency, hypertrichosis, short stature, hypotonia, distinctive facial features (thick eyebrows, long eyelashes, narrow palpebral fissures, broad nasal tip, down slanting palpebral fissures), skeletal abnormalities (clinodactyly, brachydactyly, accelerated skeletal maturation), feeding problems and behavioral difficulties (Figure 4A) (Table 2) [23,24,25]. *KMT2A* variants are distributed throughout the gene, with a pathogenic mutation hotspot in exon 27, and most of them lead to *KMT2A* loss of function. WDSTS patients usually present de novo private mutations, and the diagnosis is based on clinical evaluation of signs and symptoms then confirmed by molecular analysis. Unfortunately, a specific treatment is not available, thus possible interventions aim at reducing the severity of symptoms.

### 2.2. Other Chromatinopathies

Mutations in *KMT2A* have been also found in patients with a clinical presentation suggestive of other chromatinopathies but negative for alterations in the related known-causative genes. Their clinical presentation shares with WDSTS some phenotypic features and it is caused by alterations of genes involved in the regulation and maintenance of chromatin state as *KMT2A.* Indeed, these syndromes are caused by mutations in genes of the epigenetic machinery and therefore are known as chromatinopathies [16,18].

In 2015, a whole exome sequencing (WES) analysis of a cohort of 46 individuals with an initial diagnosis of Coffin–Siris syndromes (CSS1, OMIM #135900; CSS2, OMIM #614607; CSS3, OMIM #614608; CSS4, OMIM #614609; CSS5, OMIM #616938; CSS6, OMIM #617808; CSS7, OMIM #618027; CSS8, OMIM #618362; CSS9, OMIM #615866; CSS10, OMIM #618506; CSS11, OMIM #618779; CSS12, OMIM #619325) or Nicolaides–Baraitser syndromes (NCBRS, OMIM #601358) revealed a heterozygous de novo missense mutation in the *KMT2A* gene in a boy clinically diagnosed with CSS1 [29]. Coffin–Siris syndrome is a rare multisystemic congenital syndrome characterized by developmental or cognitive delay (from mild to severe), congenital anomalies involving different systems such as the genitourinary, cardiac or gastrointestinal (GI)systems or the central nervous system (CNS), distinctive facial features and musculoskeletal anomalies (aplasia or hypoplasia of the distal phalanx or nail of the fifth and additional digits) (Figure 4B) [30]. CSS is caused by mutations in genes encoding subunits of the BAF (ATP-dependent BRG1/BRM associated factor) complex, which functions as a chromatin remodeling factor and includes *ARID1B* (6q25.3, OMIM #614556; associated with CSS1), *ARID1A* (1p36.1-p35, OMIM #603024; associated with CSS2), *SMARCB1* (22q11.23, OMIM #601607; associated with CSS3), *SMARCA4* (19p13.3, OMIM #603254; associated with CSS4), *SMARCE1* (17q21.2, OMIM #603111; associated with CSS5), *ARID2* (12q12, OMIM #609539; associated with CSS6), *DFP* (11q13.1, OMIM #601671; associated with CSS7), *SMARCC2* (12q13.2, OMIM #601734; associated with CSS8), *SOX11* (2p15.2; OMIM #600898; associated with CSS9), *SOX4* (6p22.3, OMIM #184430; associated with CSS10), *SMARCD1* (12q13.12, OMIM #601735; associated with CSS11) and *BICRA* (19q13.33, OMIM #605690; associated with CSS12). The CSS1 patient with *KMT2A* mutation showed cardiac anomalies (patent ductus arteriosus and mitral valve prolapse), right retinal atrophy and unilateral cryptorchidism. He exhibited speech delay and peculiar signs such as bilateral fifth finger clinodactyly and dysmorphisms, including upslanted palpebral fissures, long eyelashes, a bulbous nasal tip, long philtrum and a full lower vermillion. In addition, he suffered from recurrent pulmonary infection [29] (Table 2).

Thanks to targeted sequencing and genome-wide DNA methylation analyses, in 2017, Sobreira and colleagues, investigating a cohort of 27 patients with a clinical diagnosis of Kabuki syndrome (KS1, OMIM #147920; KS2, OMIM #300867), found two patients positive for mutations in *KMT2A* (a de novo heterozygous missense mutation in pt#KS8 and a donor splice site mutation in pt#KS29) [31]. Kabuki syndrome is a congenital disease with a broad and variable spectrum, characterized by mild-to-moderate cognitive disability, post-natal growth deficit, characteristic facial features (long eyelid cracks with slight ectropion of lateral third of the lower eyelid), skeletal abnormalities and immunodeficiency (Figure 4C) [32]. In about 60% of KS cases, the syndrome is caused by mutations in *KMT2D* (12q13.12, OMIM #602113; associated with KSS1), also known as *MLL2*, while in a few cases the causative mutation is carried by the *KDM6A* gene (Xp11.3, OMIM #300128; associated with KSS2). *KMT2D* is a methyltransferase that plays crucial roles in development, differentiation, metabolism, and tumor suppression [33]. Both patients analysed by Sobreira and colleagues presented hypotonia, persistent fetal fingerpads, eversion of the lower lateral lid and long palpebral fissure; patient #KS8 in addition showed seizures and recurrent infection and brachydactyly, while patient #KS29 presented ID and feeding difficulties (Table 2) [31].

In two different works, patients with Cornelia De Lange syndrome (CdLS1, OMIM #122470; CdLS2, OMIM #300590; CdLS3, OMIM #610759; CdLS4, OMIM #614701; CdLS5, OMIM #300882)-like phenotype were found carriers of pathogenetic variants in *KMT2A*. Cornelia De Lange syndrome is a rare and clinically variable neurodevelopmental disorder characterized by ID, distinctive facial features, prenatal and postnatal growth retardation, congenital anomalies (malformations of the upper limbs, gastrointestinal malformation/rotation, heart defects and genitourinary malformations), and behavioral problems (Figure 4D) [34]. So far, mutations leading to CdLS have been identified in seven genes: *NIPBL* (5p13.2, OMIM #608667; associated with CdLS1), *SMC1A* (Xp11.22, OMIM #300040; associated with CdLS2), *SMC3* (10q25.2, OMIM #606062; associated with CdLS3), *RAD21* (8q24.11, OMIM #606462; associated with CdLS4), *HDAC8* (Xq13.1, OMIM #300269; associated with CdLS5) and the two more recently described *BRD4* (19p13.12) and *ANKRD11* (16q24.3). All these genes belong to the cohesin complex involved in chromosome segregation, DNA repair and gene regulation [34]. CdLS5 belongs to the chromatinopathies group, as HDAC8 is a histone deacetylase acting on the chromatin structure with a transcriptional repression effect. In 2015, a WES analysis on 32 Turkish individuals revealed the presence of a de novo heterozygous nonsense *KMT2A* mutation in one female patient [35]. She presented developmental and growth delay, microcephaly, clinodactyly, hirsutism, DD/ID and facial dysmorphisms including long philtrum, thin and arched eyebrows, synophrys, long eyelashes, a thin upper lip, and a high arched palate (Table 2). Two years later, Parenti and colleagues identified a de novo nonsense mutation in *KMT2A* in a male patient with CdLS clinical diagnosis, through targeted next-generation sequencing (NGS) analysis [36]. He exhibited growth retardation, mild ID and peculiar dysmorphisms (arched eyebrows with synophrys, long eyelashes, ptosis, bulbous nasal tip and thin upper vermillion border of the lip), while minor anomalies included small hands and clinodactyly of the fifth finger (Table 2).

More recently, two studies identified pathogenetic variants in *KMT2A* in patients with initial diagnosis of Rubinstein–Taybi syndrome (RSTS1, OMIM #180849; RSTS2; OMIM #613684) [26,37]. RSTS is a highly rare autosomal dominant genetic disorder, characterized by typical facial features, skeletal abnormalities (microcephaly, broad thumbs and first toes), ID, speech delay, and postnatal growth retardation (Figure 4E) [28,38,39]. RSTS is mostly caused (70%) by heterozygous pathogenic variants in *CREBBP* (16p13.3, OMIM # 600140; associated with RSTS1), or in few cases (10%) in *EP300* (22q13.2, OMIM #602700; associated with RSTS2) [40,41]. *CREBBP* and *EP300* encode, respectively, for CBP and p300, two lysin acetyltransferases (KATs) involved in the opening of the chromatin, in consequent transcriptional regulation and fundamental biological pathways [42,43,44,45,46]. In 2019, a WES analysis was carried out on patients clinically diagnosed with RSTS, and one male patient (number #103) was found positive for mutation in *KMT2A*. He presented hypotonia, skeletal anomalies involving hands and feet, and different facial features such as synophrys, arched and thick eyebrows, and downslanting palpebral fissure among others (Table 2) [37]. In 2021, a study individuated heterozygous variants in *KMT2A* in six patients with an RSTS-like phenotype but negative for RSTS known causative genes, thanks to the NGS approach (multigene panel sequencing and WES) [26]. The most common features displayed by these patients were ID (6/6), long eyelashes (6/6), speech delay (5/6), broad halluces (5/6), columella below the alae nasi (5/6), wide nasal bridge (4/6), ptosis (4/6), downslanting palpebral fissures (4/6), thick eyebrow (4/6), postnatal growth retardation (4/6) and behavioral problems (4/6). In addition, half of patients presented hirsutism and two of them showed hypotonia and feeding problems (Table 2). Importantly, abundant evidence suggests a clinical and molecular overlap for mutations in genes encoding proteins involved in the regulation and maintenance of the chromatin state. Thus, it is possible to hypothesize a thorough molecular evaluation for shared altered pathways in the future, for a correct diagnosis.

## 3. *KMT2A* Somatic Mutations

*KMT2A* somatic mutations are implicated in several tumors. The most common types of alterations involving *KMT2A* are mutations (3.62%), fusions (0.13%) (with more than 80 different partners identified) [47], losses (0.10%), amplifications (0.07%), and *KMT2A-EP300* fusions (0.19%) [48]. Among the mutations, the most frequent observed in patient-derived samples are missense (54.36%), synonymous (13.61%) and nonsense substitutions (7.34%) [49]. On the contrary, in patients with germline *KMT2A* mutations, the ones more represented are frameshift (41%) and stop mutations (29%), followed by missense variants (18%) [26]. The project GENIE, led by the American Association for Cancer Research, highlighted how *KMT2A* is implicated in many diseases, especially in blood cancers such as acute myeloid leukemia (2.49%), T-cell lymphoblastic leukemia (5.63%) and up to 14% in high grade B-cell lymphoma (Figure 5). *KMT2A* is altered also in 4.65% of malignant solid tumors, such as lung adenocarcinoma, colon adenocarcinoma and bladder urothelial carcinoma (Figure 5). Interestingly, *KMT2A* is not the only gene of the epigenetic apparatus whose somatic mutations give rise to tumors, especially in blood malignancies. In fact, somatic alterations in other chromatinopathies genes were found in myelodysplastic syndromes (*ASXL1*, *ATRX*, *DNMT3A*, *EED*, *EZH2*, *KDM6A*, *KMT2* family genes, *PHF6*) [50], acute myeloid leukemia (*ASXL1*, *DNMT3A*, *PHF6*) [51], multiple myeloma (*KDM6A*, *KMT2B*, *KMT2C*, *WHSC1*) [52] and lymphoid malignancies such as acute lymphoblastic leukemia (*CREBBP*, *DNMT3A*, *EP300*, *EED*, *EZH2*, *PHF6*) and diffuse large B-cell lymphoma (*CREBBP/EP300*, *EZH2*, *KMT2C/D*) [52,53].

## 4. Effects of *KMT2A* Mutations in Animal Models

*KMT2A* is an evolutionary conserved gene, involved in several functional process of embryonic development, ranging from hematopoiesis to neurogenesis. Indeed, in 1995, Yu and colleagues showed that the complete disruption of *KMT2A* was embryonic lethal in mice, and heterozygous animals were anemic and affected by growth delay, hematopoietic anomalies and skeletal malformations [54]. Developmental defects were investigated in *Drosophila melanogaster* too, where mutations in *KMT2A* homolog (*trx*) led to a wide range of homeotic transformations [55]. Interestingly, *KMT2A* was demonstrated as having an important role in the maintenance of memory Th2 cell function [56] and in hematopoiesis, as its absence caused defects both in self-renewal of murine hematopoietic stem cells and in hematopoietic progenitor cell differentiation in zebrafish [57,58]. In addition, impairments in neural development were observed knocking down *Kmt2a* in zebrafish, and in murine models *Mll1* was identified as a crucial component in memory formation, complex behaviors and synaptic plasticity [59,60,61,62,63].

Thus, *KMT2A*-depleted animal models recapitulate phenotypes described for patients with both germline and somatic mutations. *KMT2A* associated syndromes show clinical signs such as ID, behavioral problems, speech and growth delay and peculiar dysmorphisms, while the most frequent tumors enriched in *KMT2A* mutations are the hematological ones (e.g., B-cell lymphoma, T-cell lymphoblastic leukemia, acute myeloid leukemia), according to neurodevelopmental and hematopoietic defects found in the aforementioned in vivo models.

## 5. Epigenetic Strategies for Pharmacological Approaches

Targeting the regulators of lysine methylation is an emerging strategy for therapeutic approaches, given the role of chromatin post translational modification in regulating gene expression, and considering that lysine methylation has a pivotal role in this process. Indeed, mutations in one of the components of the epigenetic machinery affect the normal pattern of covalent histone modifications, leading to an incorrect gene expression pattern that may consequently result in tumor evolution. In addition, given the very high specificity of each methyltransferase to its target, the development of drugs directed to those enzymes would have the advantage to minimize the off-target effects [64].

As described above, *KMT2A* alterations have been reported in several blood cancers such as mixed-lineage, acute lymphoblastic and acute myeloid leukemia [65]. Acute leukemia with rearrangements of the *KMT2A* gene (KMT2Ar) is associated with a higher risk of relapse and is more resistant to standard therapies. KMT2A exerts its function by forming a core-complex with other proteins [66]; for this reason, the inhibition of KMT2A with its interaction partners, both histone and non-histone proteins, is a promising pharmacological strategy when KMT2A rearrangements are drivers of pathology, such as in leukemia. For example, recent studies have shown that the use of peptidomimetics disrupting the interaction between KMT2A and WDR5 (a member of the above-mentioned core-complex) in murine cell line reduces the expression of target genes responsible for KMT2A-mediated leukemogenesis and inhibits the growth of leukemia cells [67,68].

Similarly, it was demonstrated that the small molecule EPZ-5676 has a modest clinical activity reducing the proliferation of MLL-rearranged cells and inducing apoptosis by targeting the enzymatic core of DOT1L, a H3K79 methyltransferase recruited to fusion partners of KMT2A in disease-linked translocations and required for leukemogenesis [69,70,71,72]. Advances in treating MLL-rearranged leukemia were also achieved by using small molecules to block the KMT2A binding site on Menin, a protein encoded by *MEN1* and required for oncogenic transformation, leading to the inhibition of the aberrant leukemogenic transcription program [73,74,75,76,77].

Another pharmacological efficient approach in cancer treatment might be the targeting of pathways deregulated in tumorigenesis. Indeed, the inhibition of glycogen synthase kinase 3 (GSK3) can induce the growth arrest of leukemia cells in KMT2Ar leukemia [78], while targeting the DNA damage response (DDR) pathway can lead to specific synthetic lethality in leukemic cells with MLL-rearrangements [79].

Besides the leukemia treatment, the KMT inhibitors are considered potential drugs for other cancers. In particular, Tazemetostat has been approved in January 2020 for the treatment of a rare tumor, epithelioid sarcoma, and then for follicular lymphoma, sustaining the role of the lysine methylation pathways as potential effective targets for treating various diseases [80].

On the contrary, in genetic disorders related to *KMT2A*, the altered histone methylation status is mainly attributed to loss of functions mutations or missense mutations involving this gene. For this reason, a possible pharmacological approach could counteract the lack of KMT2A activity.

Altered epigenetic control of gene expression may cause psychosis and other psychiatric diseases, it was demonstrated that the atypical antipsychotic clozapine can induce the methylation of GABAergic gene promoters through Mll1 recruitment in a mouse model of schizophrenia [81,82]. Moreover, a study comparing clozapine-responder and non-responder twins demonstrated that clozapine increases DNA methylation of the *MECP2* promoter, leading to its downregulation, and consequently enhancing the expression of genes that are regulated by MeCP2 protein [83]. Similarly, the antidepressant phenelzine and its analogue bizine enhance H3K4me2 status in H460, A549 and MDA-MB-231 cancer cell lines by inhibiting the activity of the histone demethylase LSD1 [84]. Furthermore, tranylcypromine (TCP), another antidepressant, has been demonstrated to specifically inhibit LSD1, and its administration in combination with all-trans-retinoic-acid (ATRA) induces the differentiation of acute promyelocytic leukemia (APL) and acute myeloid leukemia (AML) blasts [85]. Moreover, a phase I/II trial (ClinicalTrials.gov: NCT02261779) have demonstrated that TCP-ATRA combined therapy can be used to treat refractory or relapsed AML patients, even if the required high dosage and the prolonged treatment may cause the onset of several side-effects [86]. For this reason, a selective LSD1 inhibitor, ORY-1001, has been developed using TCP structure. Sub-nanomolar doses of this molecule reduce the proliferation of MLL-translocated leukemic cell lines, both in vitro and in vivo, and display synergistic action with the common anti-leukemic drugs, opening the possibility of a targeted and personalized therapy [87]. A phase I/IIa clinical trial has already evaluated the tolerability, pharmacokinetics and pharmacodynamics of ORY-1001 in relapsed or acute refractory leukemia (EUDRACT no.2013-002447-29) [88].

Interestingly, epigenetic interventions could be either pharmaceutical or nutritional. It is well known that dynamic crosstalk between gut microbiota and the host is present and that it can be modulated by diet. Krautkramer and colleagues reported that microbiota regulates histone methylation and acetylation in different tissues as a diet-dependent process [89] and, notably, a microbiota-dependent epigenetic signature was reported in specific diseases, e.g., inflammatory bowel disease [90]. Indeed, the microbial community within the intestine can produce metabolites such as short-chain fatty acids (SCFAs) with a known role of histone deacetylase (HDAC) inhibitors. These compounds or diets able to increase them were recently used as possible therapeutic approach for several diseases, including drug-resistant epilepsy [91,92], cancer [93], neurodegenerative disease [94], heart failure [95], and diabetes mellitus [96], and their effect was even studied in experimental models of chromatinopathies, i.e., Kabuki syndrome [97] and Rubinstein–Taybi syndrome [98]. Furthermore, bacteria synthetize essential vitamins, fundamental for immune systems, such as B12, but also folate, required for DNA, histone and protein methylation [99,100]. Intriguingly, in a kdm5-deficient *Drosophila* model, not only an increase in gut H3K4me3 but also the disruption of intestinal barrier together with aberrant immune activation and anomalies in social behavior were observed. All these changes correlated with alterations in gut microbiota composition, which were rescued by probiotic administration [101].

Thus, considering the latest developments on epigenetic intervention, a deepening understanding of microbiota composition of patients with *KMT2A* mutations could help new therapeutic approaches investigation among the epigenetic treatments.

## 6. Final Remarks

Epigenetic modifications are fundamental for many biological processes; indeed, alterations of genes with this activity can lead to neurodevelopmental disorders or tumorigenesis, when germinal or somatic mutations respectively occur [102,103]. This is the case of *KMT2A*, a lysin methyltransferase-coding gene, whose variants are associated with a chromatinopathy (WDSTS) at germinal level or can be found in both blood cancers and solid tumors in regard to malignancies.

Interestingly, due to exome- and genome-wide analyses, patients described above with a defined initial chromatinopathy diagnosis but lacking the molecular one were found to be carriers of pathogenetic variants in the *KMT2A* gene and could have obtained a clinical re-evaluation. In detail, nearly the totality of patients previously diagnosed with CdLS, CSS, KS and RSTS showed features common to WDSTS, such as ID (11/12), speech delay (7/10), peculiar dysmorphisms affecting eyes (12/12) (i.e., thick eyebrows, synophrys, long eyelashes, ptosis and downslanting/narrow palpebral fissure) and nose (12/12) (i.e., depressed nasal bridge and broad nasal tip), while about half of them shared with WDSTS feeding problems (5/10), hirsutism (6/10) and hypotonia (6/10). Oddly, almost all of these patients displayed features less frequently present in WDSTS, such as dysmorphisms affecting mouth (7/12) (i.e., high arched palate and thin upper vermilion) and anomalies of hands/feet (11/12) (i.e., clinodactyly, brachydactyly, persistent fetal fingerpads and broad halluces). Indeed, mutations in different genes involved in the regulation and maintenance of chromatin state can lead to a clinical overlapping phenotype, suggesting a common affected pathway during embryonic development and the evaluation of an expanded set of genes when investigating the molecular causes for a correct diagnosis of these syndromes.

In addition, somatic mutations in *KMT2A* have been reported in different tumors, as well as alterations in all *KMT2* family genes [104] and in other genes associated to chromatinopathies. Curiously, we observed that germline mutations described in the literature are more frequently nonsense than missense, in contrast to somatic ones. This could be explained by the consequent loss of function mechanism characterizing most of chromatinopathies due to a defective protein production, which strongly impacts on embryonic development.

To conclude, since molecular defects in KMT2A also characterize some types of tumors, and research in the field of epigenetic drugs for malignancies is rapidly evolving [101], a therapeutic approach targeting KMT2A interaction or its pathway could be considered also for chromatinopathies, modulating epigenetic dysfunction with pharmaceutical products or diet-based interventions.

## Figures and Tables

**Figure 1 genes-13-00514-f001:**
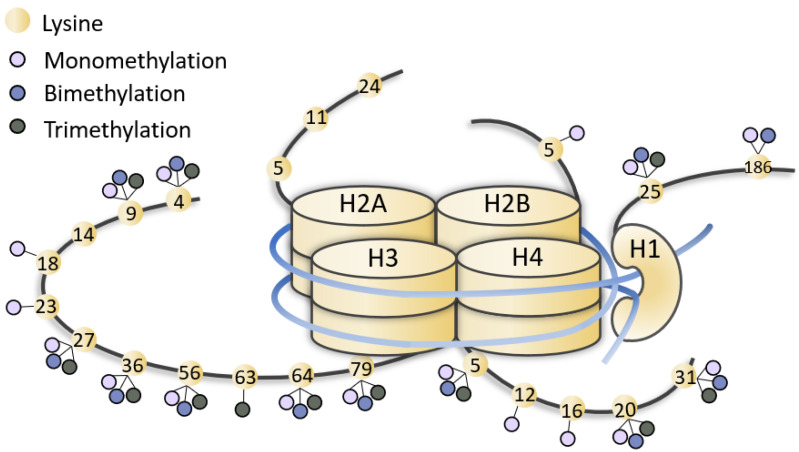
Representation of methylated lysines of histone tails. Lysines (yellow dots) of histone (H1, H2, H3, H4) tails can be mono-, bi-, tri-methylated (little lilac, blue and grey dots). The figure is not drawn to scale.

**Figure 2 genes-13-00514-f002:**
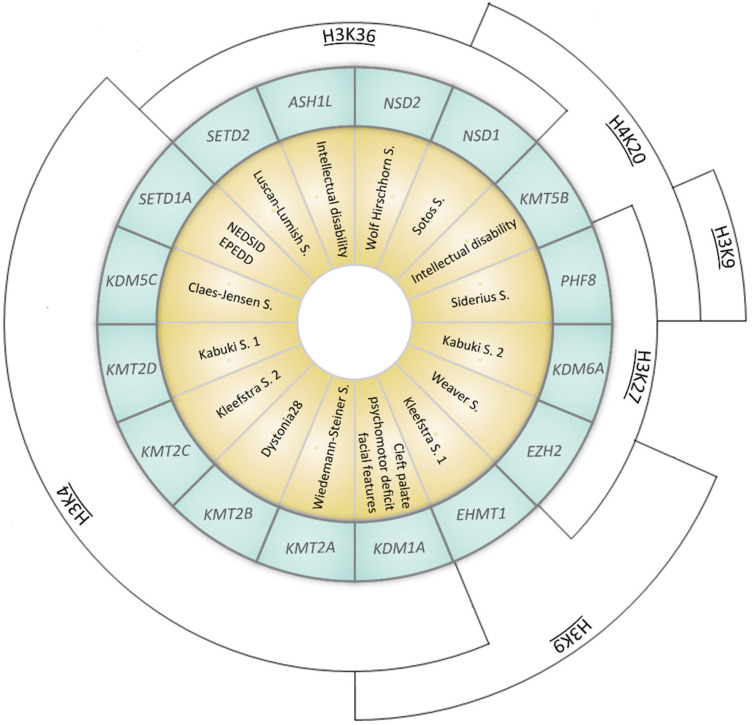
Representation of syndromes caused by mutations in genes coding for KMTs (Lysine methyltransferases) or KDMs (Lysisne demethylases). Syndromes (yellow inner ring) and the corresponding causative gene (coding for KMTs or KDMs, listed in the middle blue ring) are represented. The outer arcs indicate the site of epigenetic modification (NEDSID: Neurodevelopmental disorder with speech impairment and dysmorphic facies; EPEDD: Epilepsy, early-onset, with or without developmental delay).

**Figure 3 genes-13-00514-f003:**
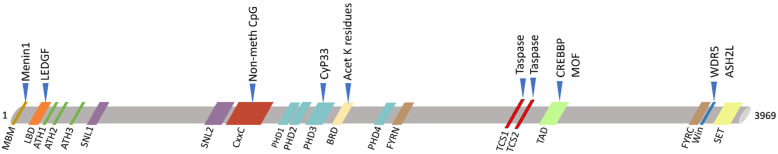
Schematic view of KMT2A protein domains (below) and its main interactors (upper). KMT2A domains: MBM, high-affinity Menin-binding motif, residues 6–10; LBD, LEDGF-binding domain, residues 109–153; ATH1-2-3, AT-Hook1/2/3, residue 169–180, residues 217–227, residue 301–309; SNL1-2, nuclear-localization signal 1/2, residues 400–443, residues 1008–1106; CxxC, including: pre-CxxC region, residues 1149–1154, CxxC domain, residues 1147–1242, post-CxxC residues 1298–1337; PHD1-2-3-4, plant homology domain 1/2/3/4, residues 1431–1482, residues 1479–1533, residues 1566–1627, residues 1931–1978; BRD, bromodomain, residues 1703–1748; FYRN, FY-rich N-terminal domain, residues 2018–2074; TAD, transactivator domain, residues 2829–2883; FYRC, FY-rich C-terminal domain, residues 3666–3747; Win, WDR5 interaction motif, residues 3762–3773; SET, Su(Var)3-9 enhancer-of-zeste trithorax domain, residues 3829–2945. KMT2A has two sites for cutting by Taspase1: TCS1-2, taspase1 cleavage site 1/2, residue 2666–2670 and residues 2718–2722.

**Figure 4 genes-13-00514-f004:**
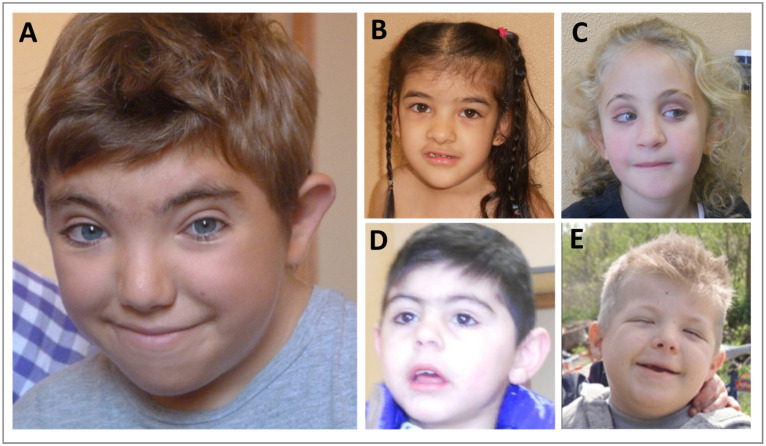
Comparison of typical facial features of (**A**) Wiedemann–Steiner syndrome [26]; (**B**) Coffin–Siris syndrome; (**C**) Kabuki syndrome; (**D**) Cornelia De Lange syndrome [27]; (**E**) Rubinstein–Taybi syndrome [28].

**Figure 5 genes-13-00514-f005:**
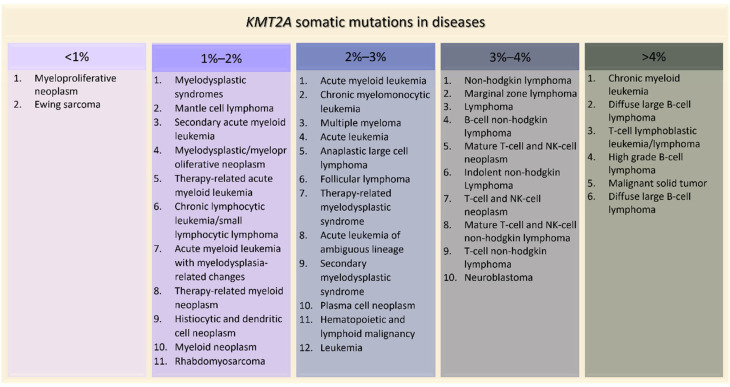
*KMT2A* somatic mutations in tumors ordered by percentage of positive cases (AACR Project GENIE).

**Table 1 genes-13-00514-t001:** Details of genes and syndromes represented in Figure 2.

Gene (OMIM *)	Associated Developmental Disorder(s) (OMIM #)	Targeted Lysine Residue
*SETD1A* (611052)	Neurodevelopmental disorder with speech impairment and dysmorphic facies NEDSID (619056)/Epilepsy, early-onset, with or without developmental delay EPEDD (618832)	H3K4 (met)
*SETD2* (612778)	Luscan-Lumish S. (616831)	H3K36 (met)
*KDM1A* (609132)	Cleft palate, psychomotor retardation, distinctive facial features (616728)	H3K4 (demet) H3K9 (demet)
*KDM5C* (314690)	Claes-Jensen S. (300534)	H3K4 (demet)
*KDM6A* (300128)	Kabuki S. 2 (300867)	H3K27 (demet)
*KMT2A* (159555)	Wiedemann–Steiner S. (605130)	H3K4 (met)
*KMT2B* (606834)	Dystonia 28 (617284)	H3K4 (met)
*KMT2C* (606833)	Kleefstra S. 2 (617768)	H3K4 (met)
*KMT2D* (602113)	Kabuki S. 1 (147920)	H3K4 (met)
*KMT5B* (610881)	Intellectual disability (617788)	H4K20 (met)
*EZH2* (601573)	Weaver S. (277590)	H3K9 (met) H3K27 (met)
*EHMT1* (607001)	Kleefstra S. 1 (610253)	H3K9 (met)
*ASH1L* (607999)	Intellectual disability (617796)	H3K36 (met)
*NSD1* (606681)	Sotos S. (117550)	H3K36 (met) H4K20 (met)
*NSD2* (602952)	Wolf Hirschhorn S. (194190)	H3K36 (met)
*PHF8* (300560)	Siderius S. (300263)	H3K9 (demet) H3K27 (demet) H4K20 (demet)

*: Gene, #: Associated Developmental Disorder(s).

**Table 2 genes-13-00514-t002:** Clinical signs reported in patients with a *KMT2A* mutation and an initial clinical diagnosis of chromatinopathy. Presence of all features is compared with the one in WDSTS.

	WDSTS	CdLS	CSS	KS	RSTS
	[26]	1 + 1 pt [35,36]	1 pt [29]	2 pt [31]	1 + 6 pt [26,37]
Vision problems	−	0/2	1/1	1/2	1/7
Cardiac problems	+	1/2	1/1	1/2	0/7
CNS problems	+/−	1/2	0/1	NA	0/7
Genitourinary problems	−	0/2	1/1	1/2	2/7
Feeding problems	+	0/2	1/1	1/2	3/7
Behavior problems	+	1/2	0/1	NA	3/7
Frequent infection	−	0/2	1/1	1/2	0/7
Seizures	+/−	0/2	0/1	1/2	1/7
ID	++	2/2	1/1	1/2	7/7
Speech delay	++	1/2	1/1	NA	5/7
Microcephaly	−	2/2	NA	NA	3/7
Eyes anomalies (thick eyebrows, synophrys, long eyelashes, ptosis, downslanting/narrow palpebral fissure)	+	2/2	1/1	2/2	7/7
Nose anomalies (depressed nasal bridge, broad nasal tip)	+	2/2	1/1	2/2	7/7
Mouth anomalies (high arched palate, thin upper vermilion)	+/−	2/2	1/1	0/2	4/7
Hands/feet anomalies (clinodactyly, brachydactyly, persistent fetal finger pads, broad halluces)	+/−	2/2	1/1	2/2	6/7
Delayed bone age	+	0/2	NA	NA	0/7
Hirsutism	+	1/2	1/1	NA	4/7
Hypotonia	++	NA	1/1	2/2	3/7

WDSTS: Wiedemann–Steiner syndrome, CdLS: Cornelia De Lange syndrome, CSS: Coffin–Siris syndromes, KS: Kabuki syndrome, RSTS: Rubinstein–Taybi syndrome, CNS: Central nervous system, ID: intellectual disability. ++ = 70–100% WDSTS patients; + = 20–70% WDSTS patients; +/− = 5–20% WDSTS patients; − = <5% WDSTS patients; NA = not assessed.

## Data Availability

Not applicable.

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
