# Peer review of "KMT2A: Umbrella Gene for Multiple Diseases"

_genes, 2022, doi:10.3390/genes13030514_

Round 1
Reviewer 1 Report
The manuscript by Castiglioni and colleagues review the structure and function of KMT2A as well as different implications in pathology, both in Mendelian genetic diseases and in cancers.
In the first part, the authors detail the Wiedeman Steiner syndrome, caused by mutations of KMT2A. They then review the literature and describe the different patients mutated in KMT2A and presenting phenotypic overlaps with other chromatinopathies.
In the second part, the implication of somatic mutations in KMT2A is discussed, particularly in blood cancers.
Finally, the third part reviews the animal models of KMT2A before discussing the various therapeutic approaches, particularly epigenetic ones, that are possible.
In general, this review is well written and easy to read, including an interesting discussion of microbiota and diet therapy approaches.
I have some comments and suggestions:
- In Figure 2 et Table 1, it is mentioned for NSD1 Sotos and Beckwith Wiedemann syndrome. Mutations in NSD1 are not classically associated with Beckwith Wiedeman syndrome but rather as a differential diagnosis of Sotos.
Similarly, SETD1a is not directly involved in schizophrenia. The OMIM Schizophrenia (181500) does not list SETD1A gene and on the other hand SETD1A (611052) is only associated on OMIM with Epilepsy, early-onset, with or without developmental delay (618832) and Neurodevelopmental disorder with speech impairment and dysmorphic facies (619056).
- Figure 1 provides little information compared to the main message of the manuscript which discusses more about the structure and functions of KMT2A.
As such, I would have replaced this figure by a figure showing the structure of KMT2A with its different protein domains as well as its different interactions with its partners and functions.
- Figure 2 et Table 1 are redundant in providing the same information. The only contribution of Table 1 compared to Figure 2 is to specify the OMIM numbers of genes and syndromes.
- On the other hand, in addition to Table 2, it might be useful to add a figure with pictures of different patients with phenotypic overlap to be able to judge the differences from WDSTS and the relevance of the initial clinical diagnosisIn the iNeurons transcriptomic data, have you highlighted other NDDs with a more concordant phenotype?
Minor comments:
- OMIM numbers for Cornelia de Lange syndrome should be specified on line 146 rather than 158
- Line 189: specify the percentage of CREBBP involvement in the RSTS
- Figure 3
- Line 265: Can you specify which study model was used and which type of molecule was used?
- Can you specify in the legend the references used to adapt the table ?
Reviewer 2 Report
The manuscript authored by Castiglioni et al., entitled “KMT2A: umbrella gene for multiple diseases” describes pathologies associated with mutations in KMT2A, a histone methyltransferase that methylates histone H3 at lysine 4. After an introductory part, the Authors focus on the occurrence and type of KMT2A mutations/alterations in various chromatinopathies and cancers and on the effects of KMT2A alterations in animal models. Then they describe pharmacological and nutritional approaches that could be used to ease the consequences of germline and somatic KMT2A mutations.
In my opinion the text, especially the introduction, is rather chaotic and does not provide necessary information to enable the Reader to see the importance of KMT2A in chromatin organization and transcription regulation. Also, some parts of the text seem too laconic (e.g. KMT2A structure and mode of action) or too extensive (description of patients symptoms). There is also a number of wording/typing errors.
Major issues:
- Lines 55-70: in my opinion this part, containing general information on KMTs, should precede the information on KMT2A.
- Lines 44-51: the part on KMT2A structure, catalytic activity etc. should be more elaborated. E.g. the KMT2 family should be mentioned, the role of the mentioned domains (e.g. bromodomain ) should be described, also whether KMT2A preferably operates on gene promoters or enhancers. It should be also mentioned that KMT2A acts in complexes with other known protein partners; without this information the paragraph 259-269 is incomprehensible. The additional information would help the Reader to better perceive the importance of KMT2A and possible consequences of its dysfunction.
- It would be better to define each chromatinopathy and its symptoms before description of particular patient cases with KMT2A mutations. Also, in my opinion, there is no need to describe the patient symptoms in such detail; rather, the Authors should, based on the info in Tab. 2, focus on these symptoms which are similar to those of WDSTS and dissimilar to those of the diagnosed chromatinopathy and, if possible, offer their opinion on the accuracy of diagnosis. In that sense I would suggest to transfer the information contained in the final remarks (L.345-360) to this part of the text.
Minor issues ( examples):
-Line 25: Authors should stick to one term (hemo- or hematopoiesis).
-line 41 -Line 59: “electric” rather than “electronic”
-line 44: (fragments) which heterodimerize and regulate
-line 66: “altered” rather than “wrong” conformation
-line 77: there are “arcs” rather than “arrows” in Fig. 2
-line 90: KMT2A mutantions rather than “variants” should be used in this context
-line 116: the word “apparatus” should not be used in this context”
-lines 237-240: please, rephrase for more clarity (knocking down of KMT2A cannot be “a regulator”)
-line257: substitute by “to minimize the off-target effects”
-line 263-264: the word “binding” or “interaction” is missing.
-line 271: what are “KMT2A cells”?
-others: “suggestive of” rather than “for”, formed “of” not “by”, etc.
